# The Allosteric Communication Network in the Activation of Antithrombin by Heparin

**DOI:** 10.3390/ijms26188984

**Published:** 2025-09-15

**Authors:** Gonzalo Izaguirre

**Affiliations:** 1Insight-DNA, Oak Park, IL 60302, USA; izaguirre.g@insight-dna.net; 2College of Dentistry, University of Illinois Chicago, Chicago, IL 60612, USA

**Keywords:** proteases, serpins, hemostasis, coagulation factors, thrombosis, molecular dynamic simulations, allosteric regulation, structural biology

## Abstract

The allosteric activation of antithrombin (AT) involves a conformational shift from a native, repressed (R) to a heparin-bound, activated (AH) state. Using computational structural analysis, we identified an evolutionarily conserved allosteric communication network (ACN) comprising the residues H120, Y131, and Y166, which undergo key structural displacements during this transition. Site-directed mutagenesis of these residues markedly enhanced AT native reactivity toward FXa and reduced thermal stability, indicating their role in stabilizing the R state. These findings support a three-step “slingshot” model in which the ACN functions as a molecular lock that restrains stored conformational energy, preventing premature activation. Heparin binding disengages this lock, triggering a cascade of structural changes that propagate from the heparin-binding site (HBS) to the reactive center loop (RCL). Additional mutational analyses of residues bridging the β-sheet A (βsA) and the RCL/exosite domains revealed a delicate energetic balance involving the S380 insertion and E381–R197 salt bridge, which collectively tune the activation threshold. Molecular dynamics simulations of ACN mutants further revealed increased flexibility at both HBS and RCL domains, consistent with concerted allosteric coupling. Together, these results provide new mechanistic insights into the structural basis of AT activation and suggest avenues for engineering heparin-independent AT variants.

## 1. Introduction

Antithrombin (AT), a serpin-class protein, is a key inhibitor of procoagulant proteases and a central component of the natural anticoagulant system that preserves hemostatic balance (Appendix A). Heparin polysaccharide chains act as an essential cofactor that enhances AT’s inhibitory activity against multiple coagulation proteases [1]. Owing to the potent multi-target anticoagulant activity of the AT–heparin complex, heparin remains the most widely used anticoagulant drug. Long heparin chains simultaneously bind AT and target proteases, facilitating inhibition by bridging the transient Michaelis complex formed between the two proteins (Figure 1). Within these chains, a specific pentasaccharide sequence (H5) allosterically activates AT, enhancing its reactivity toward factors IXa and Xa (FIXa, FXa) (Appendix A). Notably, the heparin-binding site (HBS) and the reactive center loop (RCL)—which interacts with the protease—reside at opposite ends of the AT structure (Figure 2). Although the AT structural changes underlying the allosteric transition from the native activity-silent (R-repressed) to the heparin-bound activated (AH) conformation have been elucidated, the specific residues comprising the allosteric communication network (ACN) that links the HBS and RCL, as well as the operational mechanism of this network, remained incompletely defined. A detailed understanding of this allosteric mechanism at the residue level would enable the rational design of fully activated heparin-independent AT-based anticoagulants with broad specificity. Such therapeutics could potentially replace heparin and mitigate the risk of heparin-induced thrombocytopenia (HIT).

The structures of the AH–FXa and AH-FIXa Michaelis complexes revealed that only the RCL P1-R393 residue, which binds into the S1 pocket of the enzyme’s active site, and the adjacent exosite pocket, which is bound by an Arg residue located in a loop by the enzyme’s active site, engage with the proteases [2,3]. This minimal-contact interface suggests that AT has evolved to minimize its intrinsic reactivity toward proteases by limiting protein–protein interactions [4,5,6]. During the allosteric transition from the silenced (R) to the activated (AH) form, both the exosite pocket and the P1-R393 residue become accessible for reaction with FIXa or FXa [7,8,9,10].

The HBS and the RCL/exosite domains are structurally connected through the protein core, which undergoes conformational rearrangements during the R-to-AH transition [11]. In particular, the helix D (hD) and βsA domains undergo structural shifts upon heparin binding, which are transmitted to the RCL/exosite domain. This highlights the central role of the protein core in repositioning the reactive domain to enable protease recognition and inhibition [12].

With the help of comparative structural analysis, several residues at the interdomain interfaces were identified as undergoing significant positional changes between the R and AH states. The analysis of single and multiple mutations in these residues by protease inhibition reaction kinetics and thermodynamics, protein thermal stability assays, and molecular dynamic simulations (MDS) revealed that these residues are part of an ACN that functions as a molecular lock, silencing the R form and preventing activation in the absence of heparin. The disruption of this ACN by mutagenesis released the locked conformation, producing constitutively activated native AT variants (A*) that react independently of heparin. Additional mutagenesis studies at residues connecting the RCL with the body of the protein further support a mechanism in which structural signals from the core are relayed to the reactive site, enabling its activation.

## 2. Results

### 2.1. Identification of a Network of Residues That Undergo the Largest Positional Changes During the R-to-AH Transition

The x-ray crystal structures of the AT heparin-free (R) and H5-bound (AH) forms (Protein Data Bank—**PDB**—codes 1T1F [10], and 1EO3 [13], respectively) were overlapped using the software PDB Viewer 4.10 by the repeated alignment of backbone atoms to the lowest root-mean-square deviation (RMSD) (Figure 3). The displacement of the most distal side chain C atom of every core residue between the two structures was measured. Our analysis identified several residues located at domain interfaces between the HBS and exosite/RCL domains that undergo the largest displacements compared to other nearby residues (Appendix A). The residues with the largest displacements included H120 and Y131 in hD, Y166 in hE, M89 in hB, N144 in βs2A, and M17 in the N-terminal tail (Figure 3). Relative displacements between these residues also involved large distances. H120 and M17 are 8.8 Ȧ apart in R but 2.8 Ȧ in AH. Y166 and H120 are 4.7 Ȧ apart in R but 13.1 Ȧ in AH. M89 and N144 are 5.2 Ȧ apart in R but 9.3 Ȧ in AH with Y166 ending up sandwiched between them. The residue Y131 inserts its side chain into a pocket formed by residues in βs2A in R. In AH, the Y131 side chain ends up expelled from this pocket and displaced by 4 Ȧ. In the RCL, the side chain of the P1-R393 residue in R localizes into the exosite pocket formed by R235, Y253, and H319, but away from the pocket in the AH form. Except for M17, all these residues are 100% conserved in antithrombin from a list of different vertebrates [14]. Taking all these observations together, a network of residues that undergo the largest displacements in the R-to-AH transition was identified at the interfaces between major protein domains.

### 2.2. Mutations in the Identified Residues Increased Native Antithrombin Reactivity with Factor Xa

We made single valine or leucine mutations to residues M17, M89, H120, Y131, N144, and Y166. Replacing large side chains with isosteric ones like Leu or Val instead of Ala helps avoid protein instability due to the creation of large intramolecular empty spaces. The proper functionality of the mutant AT variants as canonical serpins was determined by testing their reactivity with thrombin, which is insensitive to allosteric activation. The mutations did not significantly alter the second order association rate constant for the formation of the transient Michaelis complex with thrombin (Appendix A). In contrast to thrombin, the reactivity of the native AT variants with FXa was significantly enhanced by most mutations (Figure 4). The M17 did not affect reactivity, and only the N144V mutant reacted at a slightly slower rate compared to the native wild-type AT (~60%).

The H120V mutation increased native reactivity close to the level reached by the activated wild-type AH form (236 out of the 360-fold jump difference between the native and activated wild-type AT), i.e., the native mutant protein (A*) was practically fully activated. The further addition of H5 (A*H) to the FXa inhibition reaction still increased the reactivity rate even above the level of the wild-type AH form (Figure 4), thus showing that the mechanism of activation was still operational and responsive to heparin. The double and triple mutations combining the residues H120, Y131, and Y166 activated AT, although not in an additive manner. The double Y131-Y166 mutation increased native reactivity 19-fold (A*), and adding H5 (A*H) further increased reactivity to levels above that of the wild-type AH form. Altogether, mutations in the residues H120, Y131, and Y166 seem to have altered the mechanism that keeps the R form in an activity silent state and in accordance with these residues being part of a silencing ACN.

### 2.3. Functional Cooperation Among ACN Residues

The non-additive effects of the multiple mutations that unsilenced the R form in favor of activated A* forms suggest that these residues -H120, Y131, and Y166- may cooperate to keep R stable. This hypothesis can be evaluated by site-specific thermodynamic-coupling analysis (Appendix A). The cooperative contributions of these residues to the ACN function were investigated by determining the thermodynamic linkage that the combined effects of the mutations have on the second order association rate constant for the inhibition of FXa [15]. Thermodynamic linkage is strong evidence that residues perform a common function cooperatively. Cooperation is indicated by coupling index values greater than one (Appendix A). On the other hand, the absence of thermodynamic linkage, denoted by a coupling index value of one, indicates that the effects of the mutations are additive and not cooperative. Using the rate values shown in Appendix A, we found strong thermodynamic linkage involving the residues H120, Y131, and Y166 (Figure 5).

The H120 linkage with Y166 produced the highest coupling index value. H120 was also strongly linked to Y131, as well as Y131 to Y166. Noticeably, the linkages of Y166 to H120 or Y131 were both lost when H120 and Y131 were simultaneously mutated, thus evidencing the dependence of the functional linkage between two residues on the presence of a third. These results provided further direct evidence that these residues cooperatively participate in an ACN whose function is to silence the reactivity of the R form.

### 2.4. Protein Physical Property Changes Associated with the R-to-A* Transitions

Our previous studies on the mutagenesis of the RCL P14-S380 residue showed that gains and losses in the native AT reactivity toward inhibiting FXa correlate with decrements and increments in the protein melting temperature, respectively [16]. Accordingly, it was predicted that the activated A* forms derived from mutating the ACN residues may also show decrements in protein thermal stability. Indeed, these ACN mutations led to large reductions in melting temperature, T_m_ (Figure 6).

The single residue mutation with the largest reduction in T_m_ was that of H120 followed by Y166. However, their double mutants produced a smaller decrease. It was the triple mutation, combining residues H120, Y131, and Y166, that produced the largest drop in T_m_. On the other hand, the mutation in N144 produced an increment in T_m_ and, accordingly, caused a decrement in reactivity. Only the mutations to the residues M17 and M89 produced increments in T_m_ combined with no effect or increased reactivity.

The R-to-AH transition is normally accompanied by a gain of about 40% in the protein’s tryptophane fluorescence that results from conformational shifts induced by heparin binding [17]. Changes to the protein’s baseline fluorescence and fluorescence gain induced by mutations are indicators of conformational perturbations to the AT structure. Baseline fluorescence was determined from the slope of the linear dependence of protein fluorescence on increments in the AT concentration (Appendix A). The fluorescence gain was measured by adding five times the saturating concentration of a high affinity heparin pentasaccharide to an AT solution after determining the baseline fluorescence (Appendix A). While the single mutation in H120 increased the protein’s baseline fluorescence almost two-fold, the double and triple mutations involving Y131 and Y166 decreased it (Figure 7). Noticeably, the fluorescence gain was drastically reduced or abolished by the single and multiple mutations of these three residues. The mutation in M89 also decreased baseline fluorescence and eliminated the fluorescence gain. The mutation in N144 only partially reduced the fluorescence gain. The observed alterations to the physical properties of the AT protein produced by the mutations in the ACN residues indicate conformational shifts conducive to higher reactivity that are in agreement with AT switching from the R to A* forms.

### 2.5. Transmission of the Allosteric Structural Changes to the RCL/Exosite Domain

The liberation of the RCL/exosite domain from its restrictions to allow reactivity constitutes the end result of the regulatory mechanism triggered by heparin. How the core movements are transmitted to the RCL has been a long-standing question. Electrostatic interactions around the RCL hinge region in the R form contribute to keeping the RCL P13-E381 and P14-S380 residues (N-terminal to the reactive P1-R393) inserted into the βsA gap between βs3A and βs5A [18]. In the interface between the βsA and RCL/exosite domains, these two and other residues undergo large displacements (Figure 8).

E381 is sandwiched between residues E374 (βs5A) and K222 (βs3A), and forms a salt bridge with R197, which is located on the loop at the end of hF. This salt bridge is maintained intact in the R and AH forms, which suggests that this linkage may be involved in transmitting movement to the RCL. The E381G mutation was designed to disrupt this salt bridge, and it led to an increment of the baseline fluorescence by 2.5-fold and a decrement of the fluorescence gain by 7-fold (Appendix A). The mutation also had a detrimental effect on native reactivity by decreasing the rate of inhibition of FXa by 4-fold (Appendix A), which inversely correlated with an increment in the protein’s melting temperature by 6 °C (Appendix A). This mutation also reduced pentasaccharide induced activation by 2.7-fold to a final 10-fold lower total reactivity compared to the wild-type AH (Appendix A). Furthermore, the E381G mutation abolished the enhanced native reactivity of the double H120L-Y166L mutant when combined in their triple mutant (Appendix A) and eliminated any further activation by heparin (Appendix A). Another approach to disrupt the E381-R197 salt bridge was by making the E195R-R197E double mutation that switched electrostatic charges between the two residues in the loop after hF. This electrostatic switch enhanced native reactivity 3.6-fold and 1.1-fold in the presence of H5 (Appendix A).

S380 stabilizes the RCL by its insertion in between the strands β3A and β5A (Figure 8). In a previous study, we showed that the S380G mutation increases native reactivity 3.6-fold [17]. The triple mutation combining E195R-R197E with S380G enhanced native reactivity by 72-fold, synergistically surpassing the additive effect of the two separate mutations (12-fold) (Appendix A). The melting temperature of the triple mutant decreased by 1.9 °C, and the baseline fluorescence was not perturbed (Appendix A). However, the fluorescence gain dropped 8.7-fold. The S380G mutation was also combined with the Y131L mutation to test if this combination could also produce synergistic effects on reactivity. This double mutation increased native reactivity 95-fold (Appendix A), which is comparable to the added effect by the two single mutations (31- by 3.6-fold = 112-fold), thus showing an additive-like rather than synergistic effect. Overall, these results indicate that the E381-R197 salt bridge and the S380 residue play critical roles in the allosteric mechanism by linking the RCL to the body of the protein.

### 2.6. Molecular Dynamic Simulation of the Antithrombin ACN Mutants

The dramatic effects on the AT chemical and physical properties of the mutations in the ACN residues suggest that these mutations induced significant perturbations to the protein structure. These perturbations were analyzed by computer molecular dynamic simulations (MDS). The ACN residues in the R state structure (1T1F) were mutated in silico-H120V, Y131L, and Y166V. The structures were first energy minimized to reduce steric clashes and allow for a smoother transition into MDS. The alignment of the minimized wild-type (WT) and mutant structures against their pre-minimized states produced larger RMSD values with the mutants (1.15, 1.10, and 1.04 Å, respectively) compared to the WT (0.85 Å). This difference indicates that some degree of structural destabilization was introduced by the mutations. The WT and mutant minimized structures were subjected to 1000 nano seconds (ns) of MDS to compare their progression into a more stable state. After 500 million iterations, the WT and mutant structures have followed significantly different simulation trajectories as determined by Principal Component Analysis (PCA) (Figure 9). The WT structure remained highly localized near the origin, behavior that is consistent with a stable, compact conformational state, while the mutant structures diverged sharply, exploring a broader conformational arc and eventually settling into a distinct region far from WT. This difference from the WT suggests that the mutations induced substantial conformational flexibility and a shift in equilibrium state.

The WT simulation became restricted to a collection of short exploratory trajectories, in which the Principal Component 1 (PC1) captured only ~17.4% of the total variance associated with the structural fluctuation, and PC1 to PC5 added to only ~47.5% (Appendix A). The Y166V simulation also produced multi-modal dynamics involving multiple localized movements that captured ~17.5% of the variance in PC1 and ~42.8% in up to PC5. The H120V and Y131L simulations produced a dominant collective motion that captured ~90 and 80% of the total variance in PC1, respectively. This indicates single, strong, coherent conformational fluctuation that followed directional trajectories. The resultant vectorial displacements in the PC1 versus PC2 space for the mutant trajectories were longer—about 2.4, 6.0, 4.3, respectively—than for the WT (about 0.4).

Identifying the protein regions with higher flexibility attained during the simulations required the analysis of snap shots of the structures taken at different times during the simulation progression. The protein structural changes, measured in RMSD (Å), along the simulation trajectory (ns) showed the lowest initial and further drift values for the WT compared to the mutants, especially the H120V (Appendix A). Because of this, the structural changes in the H120V protein were analyzed further to probe for the structural motifs involved in the allosteric mechanism. The changes that accumulated in the WT structure after 1000 ns of simulation, RMSD values, were subtracted from those in the H120V structure (Figure 10). The difference specifically pointed to the RCL and the HBS as hot spots of high flexibility. These results clearly show concerted communication between domains residing at opposite ends of the AT structure. Surprisingly, the region with the highest flexibility was the helix F.

Closer residue-level dynamics was analyzed by performing a Pearson correlation map based on Cα atom fluctuations of the H120V trajectory and then subtracting the map of the WT trajectory. Pearson maps reflect the collective dynamics that emerge after removing rigid-body drift. This map captures how local (on the diagonal) or distal (off the diagonal) parts of the protein move relative to one another, revealing dynamic couplings, modular organization, and potential allosteric pathways. Our differential correlation approach clearly identified the regions hF-R197, hI, βsA, and RCL as those with the highest internal dynamics affected by the mutation (Appendix A). This mapping of the differential correlated regions on the protein structure revealed the dynamic connection between the HBS and RCL through hF via the R197-E381 link as the most viable mechanism for the allosteric silencing-activation of AT (Figure 11).

The structure snap shots along the WT and mutant simulation trajectories were also analyzed for changes to the ACN—H120, Y131, and Y166—and P1-R393 residues. The side chain of the P1 residue in the crystal structure of the heparin-free AT (PDB code 1T1F) seems immobilized by being inserted into the exosite pocket (Figure 2). However, the WT simulation showed that this side chain rather fluctuates freely in a relatively immobilized RCL (Appendix A). Also, the Y131 side chain appears to have more mobility compared to H120 and Y166. These two last residues acquired mobility in some of the mutant simulations.

## 3. Discussion

The present studies on the allosteric activation of AT build upon previous structural and kinetic analyses demonstrating that the AT inhibitory reactivity toward FXa requires the allosteric, heparin-dependent activation of the RCL/exosite domain [2,7]. These past studies also established that the HBS and the RCL/exosite domains are structurally linked through core domains that undergo rearrangements upon heparin binding, resulting in significant conformational differences between the native, activity-silent (R) and heparin-bound, activated (AH) forms [11]. In the present study, through computational structural analysis, we identified an evolutionarily conserved network of residues extending from the HBS to the RCL/exosite domain, consistent with the architecture of an ACN that mediates the transmission of conformational changes. Our findings align with a previous report that mapped global structural shifts and highlighted several of the same residues [19]. The mutational analysis of residues in this network showed significant increments in the reactivity of native AT, supporting a model in which this network functions to maintain R in an activity-silent conformation. Together, these results support a mechanistic model in which the R form is stabilized through a locking system that is allosterically disengaged upon heparin binding, thereby enabling the transition toward the reactive AH state.

AT functions as a two-state conformational switch (Appendix A), in which the native A form binds heparin with significantly higher affinity than the R form. In the heparin-bound AH state, the RCL/exosite domain adopts an exposed, reactive configuration exhibiting a second-order association rate constant value of ~1 × 10^6^ M^−1^s^−1^ toward FXa. In contrast, in the R state, the RCL P1-R393 residue and the exosite are inaccessible for protease interaction, thereby suppressing reactivity [10]. The reactivity of the R form with FXa has been estimated at approximately 0.5–1 × 10^3^ M^−1^s^−1^, based on the kinetic analyses of AT variants with their RCL hinge residues P14–S380 or P13–E381 immobilized [12,20]. These values support a model in which the basal reactivity of AT in the native equilibrium mixture (R ⇄ A), measured at ~4 × 10^3^ M^−1^s^−1^, results from a population distribution heavily skewed toward the R state—approximately 99.5% R and 0.5% A. This extremely biased equilibrium reflects a stabilization energy that is remarkably high compared to that of most other proteins, underscoring the tightly controlled nature of AT’s allosteric regulation.

Achieving the native equilibrium bias (approximately 99.5% R) likely depends on strong cooperative interactions among the HBS, hD, βsA, and RCL/exosite domains. Our previous scanning mutagenesis study targeting the RCL P14–S380 residue demonstrated that changes in native reactivity correlate with shifts in protein melting temperature [16]. Specifically, shifts toward the R form resulted in increased thermal stability, while shifts favoring the A form were associated with decreased melting temperatures. This inverse relationship indicates that the R state is stabilized by interactions that are weakened or lost in the A conformation.

The results of the present study are consistent with this interpretation. Mutations of the ACN residues H120, Y131, and Y166 significantly increased native AT reactivity (A* forms) toward FXa and were accompanied by substantial reductions in melting temperature, indicating a destabilization of the R form. Additionally, proteins with mutations in these ACN residues bound stronger to immobilized heparin during purification than the WT protein, suggesting a shift in conformation toward the activated A state. These mutants also exhibited enhanced reactivity toward FXa in the presence of H5, confirming that they remained responsive to heparin activation. However, unlike the WT protein, the mutants failed to show the typical ~40% increase in tryptophan fluorescence upon heparin binding, a signal we routinely use to quantify binding affinity. This loss of the gain in fluorescence further supports the idea that these mutants exist in conformations resembling that of the AH form.

Moreover, the effects of the combined mutations on reactivity were found to be strongly thermodynamically coupled, suggesting cooperative interactions among H120, Y131, and Y166 in regulating the allosteric mechanism. Together, these findings support our hypothesis that high-energy interactions involving the ACN residues act collectively to stabilize the structurally repressed, and activity-silent R conformation.

The HBS and RCL/exosite domains are connected via the hD and βsA domains, which adopt distinct conformations in the R and AH forms, as suggested by their crystal structures. These four domains appear to operate in a modular fashion, both structurally and functionally, as it is indicated by measurements of heparin-binding affinity with AT variants whose structural linkages between the HBS and RCL/exosite domains were disrupted [21,22]. Additionally, our prior studies have shown that mutations in the RCL hinge P14–S380 residue can shift the RH–AH equilibrium by decoupling local activating conformational changes and stabilizing so-called intermediate RH forms [16]. One other crystal structure of heparin-bound AT further supports this modular model, as the structure revealed only a partial set of conformational changes typically associated with the fully activated AH form [23]. Collectively, these findings suggest that in the native R state, each domain is held in a biased equilibrium, stabilized by strong interdomain interactions. Upon allosteric activation, these interdomain constraints are relaxed, allowing each domain to sample a broader and less restricted conformational space, consistent with a more flexible AH form.

Based on the spatial positioning and displacement of the ACN residues in the R-to-AH conformational transition, along with the effects of their mutation on the AT reactivity and stability, we developed a mechanistic model that explains the role of these residues in stabilizing the R form and mediating activation upon heparin binding. This model proposes a three-step, or “slingshot”, allosteric mechanism in which the first step is the energy stored in an elastic β-sheet A where strands 3A and 5A are stretched apart. The second step is the locking or holding phase where the system is in its tensioned R state and stabilized by the residues H120, Y131, and Y166 that prevent the premature release of the stored energy. The third step is the release of the stored energy triggered by the binding of heparin that disengages the latch and propells an activating conformational shift transmitted to the RCL/exosite domain.

The latching role of the locking mechanism seems to be played by the H120-Y166, and βs2A-hD-Y131 interactions. In previous studies, we forced the elongation of hD by introducing helix stabilizing mutations, thereby demonstrating that this extension alone—along with removal of Y131—was sufficient to induce significant activation [24,25]. At the bottom of hD, H120 strenghtens the non-extended conformation of the helix by forming a stabilizing interaction with Y166, and this interaction in turn helps maintain a rigid hD-βs2A structure. The binding of heparin release the lock by forcing H120 away from Y166, therefore extenings hD and loosening the hD-βs2A interaction. Simultaneously, Y166 inserts between N144 and M89 into a secure position. The single H120V mutation showed the central role this residue plays in the locking mechanism by triggering the largest activating effect.

Our past models of the mechanism for the transmission of structural changes to the RCL were influenced by two prominent conformational changes in the R-to-AH transition revealed by crystallography: (1) the closing of the gap between β-strands 3A and 5A, and (2) the expulsion of the RCL hinge region—specifically residues P13-E381 and P14-S380—from this gap. These structural rearrangements should therefore expose the P1 residue and bring residues Y220 and K222 in βs3A into closer proximity with F372 and E374 in βs5A to form the stabilizing pairs Y220–F372 (hydrophobic) and K222–E374 (electrostatic). However, the present MDS studies suggest that the allosteric transmission of conformational changes may not require the expulsion of the RCL hinge from its insertion between β-strands 3A and 5A. However, it is important to point out that the length of the MDSs probably does not cover the whole R-to-A* structural transitions, and that adding heparin to the FXa inhibition reactions with the ACN mutants still further enhanced reactivity.

A distinctive feature of the RCL hinge is the salt bridge between the RCL P13-E381 and R197 residues. As reported by the crystal structures, this salt bridge is preserved after activation. The disruption of this bridge in the present study confirmed its critical role in mediating the allosteric communication necessary for activation. It also showed that the position of E381 is stabilized by the insertion of the RCL P14-S380 residue into a pocket between β-strands 3A and 5A. The opposing forces between R197 and S380 seem to stabilize the E381 position around a critical threshold. Our previous mutagenesis studies have shown that mutations that reinforce S380 insertion—such as those that increase side chain bulk—further stabilize the RCL hinge and reduce AT reactivity by hindering the movement of E381 [16]. In contrast, mutations like S380G increased reactivity by weakening the insertion of the P14 residue, thereby loosening the anchoring of E381 and facilitating its movement. The disruption of the E381–R197 salt bridge seems to have altered this stability threshold. For example, the E381G mutation eliminated the salt bridge and led to reduced reactivity, as was also observed in a previous study with the E381A mutation [26]. Without an anchored E381 to hold the RCL in place, S380 adopts a more deeply inserted position, thus reinforcing RCL stability and reducing reactivity. Another example is the electrostatic charge-switch mutation between E195 and R197 that appears to have redirected the E381 salt bridge from R197 to R195. Since R195 lies farther from the βs3A–βs5A gap, this shift would pull the RCL away and decrease S380 insertion, therefore, partially activating the protein. Notably, the combination of the S380G and charge-switch mutations resulted in a synergistic increase in reactivity that exceeded by far the additive effects of each mutation alone, suggesting that S380 and E381 act in a cooperative manner to regulate RCL positioning. In contrast, combining the S380G mutation with Y131L did not yield cooperative effects, indicating that Y131 operates independently from the RCL hinge. Overall, these findings underscore the importance of a precise tuning of the interplay between S380 insertion and the E381–R197 salt bridge that is essential in setting the correct response threshold for proper RCL movement and regulation of the P1-R393 residue.

The solid trajectory obtained by MDS of the H120V structure provided a good glimpse of the structural changes that allosterically link the HBS and RCL and that should lead to AT activation. In the WT simulation, the protein remained stable without proceeding into a trajectory. On the contrary, the H120V mutation unleashed a dramatic structural change consistent with the protein being in a tense and unstable state. Noticeably, high levels of flexibility were immeditely attained at both the HBS and RCL/exosite domains, suggestive of allosteric communication between the two protein domains. Changes in correlation among residues identified helix F as the most flexible secondary structure followed by the RCL, and strands 5 and 6 of βsA. The connection between hF and RCL through the E381-R197 bridge suggests that this is a critical route of allosteric communication between the HBS and RCL (Figure 12).

In conclusion, our findings delineate a comprehensive mechanistic framework for the allosteric activation of AT, revealing a strong conformational locking system that stabilizes the R state and is disengaged upon heparin binding. Central to this mechanism is an evolutionarily conserved ACN, in which residues H120, Y131, and Y166 operate cooperatively to stabilize structural constraints linking the HBS to the RCL. The three-step “slingshot” model proposed here accounts for both the structural energy storage in the R state and its triggered release upon heparin binding, ultimately promoting RCL displacement and AT activation. Mutagenesis and biophysical analyses, together with MDSs, provided strong evidence for the dynamic coupling between the HBS and RCL regions, with helix F acting as conduit for allosteric transmission through the modulation of the finely tuned position of the RCL by the S380 insertion and the E381–R197 salt bridge. These insights clarify long-standing questions about the AT’s activation mechanism and highlight critical residues and interactions that may be targeted in the therapeutic modulation of its anticoagulant function.

## 4. Materials and Methods

### 4.1. Computational Analysis of Protein Structures

The structures of the heparin-free and H5-bound AT forms were aligned using the software RCSB PDB Viewer 4.10 [27]. The crystal structures of the R (PDB code 1T1F) and AH (PDB code 1EO3) AT forms were overlayed by their backbone atoms down to an RMSD of 1.48 Ȧ. The distance of the most distal carbon atom for each residue side chain between the two structures was measured. The atomic distances are listed in Appendix A.

### 4.2. Protein Production

Recombinant AT variants were constructed on an N135Q background to block glycosylation as in our previous studies [9,12]. Site-directed mutagenesis was carried out using the Stratagene QuikChange Site-Directed Mutagenesis kit (Agilent Technologies, Santa Clara, CA, USA) and the appropriate primers. Mutations were confirmed by DNA sequencing. Variants were produced in baculovirus-infected insect cells (sf9) using the expression system from Invitrogen (Thermo Fisher Scientific, Carlsbad, CA, USA), as described by the manufacturer. AT protein variants were purified from culture supernatants by immobilizing the protein onto heparin-Sepharose followed by elution with a sodium chloride gradient. Most proteins mutated in the ACN residues bound heparin with higher-than-normal affinity and eluted at sodium chloride concentrations higher than 2 M producing preps of over 95% purity. Otherwise, preps were run over ion exchange chromatography using a mono Q column (Invitrogen, Thermo Fisher Scientific, Carlsbad, CA, USA). Those fractions containing pure antithrombin were finally concentrated and desalted by ultrafiltration, and concentrations were determined in 10% SDS-PAGE by comparison to calibrated concentrations of bovine serum albumin after staining with Coomassie Blue. Finally, proteins were stored in 100 mM Hepes buffer pH 7.4, 100 mM NaCl at −80 °C in aliquots. 

### 4.3. Baseline Protein Fluorescence Measurement

Tryptophane protein fluorescence was determined in 500–800 μL of 100 mM Hepes buffer pH 7.4, 100 mM NaCl at 25 °C by adding repeated volumes of the AT solution (2.5 μM) into a quartz cuvette. Fluorescence readings (wavelengths: excitation = 280 nm, emission = 345 nm) were recorded using a PTI Spectrofluorometer (Photon Technology International, Horiba Scientific, Piscataway, NJ, USA) and adjusted for buffer fluorescence and volume changes. The linear dependence of fluorescence increments on adjusted protein concentrations was taken as a measure of the baseline protein fluorescence.

### 4.4. Protein Thermal Denaturation

Thermal denaturation of AT variants was monitored by the temperature-induced changes in tryptophan fluorescence measured at the same conditions as described above for baseline protein fluorescence determinations. A temperature-controlled SLM 8000 (SLM Instruments, Inc. Urbana, IL, USA) spectrofluorometer, upgraded with a temperature control (Quantum Northwest, Olis, Inc., Liberty Lake, WA, USA), was used for temperature scanning in the range from 20 to 85 °C at intervals of 2 to 5 degrees (Appendix A). AT concentrations used were from 400 nM to 2 μM depending on the baseline fluorescence. The data were fitted to the Van’t Hoff equation to determine the melting temperature (T_m_).

### 4.5. Protease Inhibition Reactions

Assays of protease activity reactions with chromogenic substrates were measured as described in our previous studies [9]. Reactions were conducted at 25 °C in 0.02 M sodium phosphate buffer pH 7.4, 0.1 M NaCl, 0.1 mM EDTA, 0.1% polyethylene glycol 8000, 0.1 M S-2238 (BioMedica Diagnostics, Windsor, NS, Canada) for thrombin, and 0.1 M Hepes buffer pH 7.4, 0.1 M NaCl, 5 mM CaCl2, 0.1 mM EDTA, 0.1% polyethylene glycol 8000, 0.1 M Spectrozyme-FXa (Sekisui Diagnostics, Burlington, MA, USA) for FXa. Reactions were run in 1 mL cuvettes and the linear increase in absorbance at 405 nm was monitored using a temperature-controlled DU-640 multicell Spectrophotometer (Beckman-Coulter, Brea, CA, USA).

Second order association rate constant values for the inhibition of 3–5 nM FXa (Enzyme Research Laboratories, South Bend, IN, USA) or thrombin (U.S. Biochemical Corp, Cleveland, OH, USA) by AT variants were determined in the absence or presence of heparin pentasaccharide (Fondaparinux, Sanofi-Aventis, Toulouse, France). Reaction rates were measured under pseudo-first-order conditions by using at least 10-fold molar excess of inhibitor over protease as in previous studies [9]. Inhibition reactions (100 μL) were conducted in 1 mL cuvettes for the assigned time and terminated by adding 900 μL of protease substrate solution for measuring the residual enzymatic activity. Residual enzyme activity values at time points that span the full inhibition reaction time courses for the exponential loss of enzyme activity were fit by a single exponential function with a zero-activity end point to obtain the pseudo-first order rate value. This value was then divided by the AT concentration (0.1–1.0 μM) to obtain the second order association rate constant value. For reactions containing heparin, the pentasaccharide was present at levels that saturated AT. Rate values were corrected by factoring the stoichiometry of inhibition of the protease by the AT variants (Appendix A).

The stoichiometries for the reactions of the AT variants with thrombin and FXa were determined from reactions in which increasing concentrations of the inhibitor were added to a fixed concentration (100 nM) of protease (25–100 μL). Reactions were carried out both in the absence and in the presence of heparin pentasaccharide, which was fixed at an equimolar concentration with the inhibitor. After incubating for times sufficient to complete the reaction (95%), 5 μL of the inhibition reaction mixture were added to 1 mL of substrate (100 μM S-2238 for thrombin or 100 μM Spectrozyme-FXa for FXa), and the residual enzymatic activity was measured from the linear increase in absorbance. The decrease in protease activity with increasing molar ratio of inhibitor/protease was fit by linear regression to obtain the stoichiometry from the abscissa intercept.

### 4.6. Molecular Dynamic Simulations of Antithrombin Structures

Simulations were conducted on a virtual machine (VM) provisioned through Google Cloud Platform (GCP) using a G2-standard instance type, designed for GPU-accelerated workloads. The VM was configured with the following specifications: Instance Type: G2-standard; GPU: 1× NVIDIA L4 Tensor Core GPU; CPU: 8 vCPUs; Memory: 48 GB RAM; Operating System: Ubuntu 22.04 LTS; Storage: 200 GB persistent SSD; Zone: us-central1b.

Energy minimization of the antithrombin structure was performed using the program OpenMM 8.0. The starting structure was refined using the Amber14 force field (amber14-all.xml) with the TIP3P explicit water model (amber14/tip3p.xml). The system was constructed with periodic boundary conditions and Particle Mesh Ewald (PME) electrostatics, using a nonbonded cutoff of 1.0 nm and constraints applied to hydrogen bonds (HBonds). The structure was solvated by enclosing it in a cubic water box using Modeller, with a padding distance of 10 Å from the protein surface to the box edge and then neutralized by adding the required number of counterions (Na^+^ or Cl^−^) to achieve overall charge neutrality.

The simulation used a Langevin integrator at 310 K with a friction coefficient of 1 ps^−1^ and a time step of 2 fs. The potential energy of the system was minimized using steepest descent followed by the L-BFGS algorithm, limited to 10,000 iterations. The pre- and post-minimization structures were compared to calculate RMSD values using PyMOL software. The scripts to perform minimizations using OpenMM 8.0 in the VM and structural alignments using PyMOL in a local computer are available in the Appendix A.

Molecular dynamics simulations were performed using the OpenMM engine on the energy-minimized structures (AT_minimized.pdb). The Amber14 force field with TIP3P water model (amber14-all.xml, amber14/tip3p.xml) was used to parameterize the system. Long-range electrostatic interactions were handled using the PME method with a 1.0 nm nonbonded cutoff. Hydrogen-containing bond lengths were constrained using the HBonds setting, allowing a 2 femtosecond (fs) integration time step. The system was coupled to a Langevin thermostat at 310 K (1/ps friction coefficient) and a Monte Carlo barostat to maintain 1 atm pressure. Velocities were initialized according to the Maxwell-Boltzmann distribution at 310 K. The simulation was run for a total of 1000 nanoseconds, corresponding to 500 million steps. Snapshots of the atomic coordinates were saved into separate PDB files enabling structural analysis at discrete time points throughout the trajectory. The script is available in the Appendix A.

### 4.7. Principal Component Analysis of Antithrombin Simulations

Principal Component Analysis (PCA) was performed to compare the large-scale conformational dynamics of WT and mutant antithrombin variants over the course of 1000 ns MDS. Simulations were conducted at 2 fs timesteps, and coordinates were saved every 20 ps; therefore, each trajectory contained 50,000 frames.

Analysis focused on the protein backbone by extracting only Cα atoms from the simulation trajectories using MDTraj 1.9.9. Frames were aligned to the initial structure to remove global translational and rotational motion. The Cartesian coordinates (x, y, z) of each Cα atom were flattened per frame to form a two-dimensional matrix of shape (frames × atoms × 3), which served as input for PCA.

Dimensionality reduction was performed using the PCA implementation from scikit-learn 1.4.2 (Python 3.10). The first 20 principal components (PCs) were computed for each trajectory, and their associated variance ratios were used to assess the distribution of collective motions. Time-resolved PCA was also performed on the final 15% of each trajectory to evaluate convergence and dynamic stability.

All analyses were performed on a Google Cloud Platform virtual machine equipped with an NVIDIA A100 GPU, 24 virtual CPUs, and 96 GB RAM, running Ubuntu 22.04 LTS. Visualization of PCA results, including PC projections and explained variance trends, was performed using Matplotlib 3.8.4 and Seaborn 0.13.2. Script is available in the Appendix A.

### 4.8. RMSD Calculations

Root-mean-square deviation (RMSD) analyses were performed using PyMOL (version 2.5.10) to assess conformational changes in WT and mutant structures during MDS. Energy-minimized structures were aligned to their respective 1000 ns simulation endpoints using the align function in PyMOL 2.5.10 to ensure consistent backbone superposition. RMSD heatmaps were generated by running a custom Python script within PyMOL 2.5.10, which computes per-residue RMSD values between the energy-minimized and simulated structures. Visualizations were rendered using a red-white-blue color gradient representing increasing structural displacement (0–5 Å). For direct comparison between the WT and mutant simulations, all 1000 ns endpoint structures were aligned, and the atom wise RMSD values were calculated similarly to highlight mutation-induced structural divergence. Atom counts were verified to be equal across pairs to avoid alignment artifacts. Script is available in the Appendix A.

### 4.9. Pearson Correlation Analysis of Residue Dynamics

To quantify correlated motions between residues during MDS, Pearson correlation coefficients were computed based on the atomic displacements of Cα atoms. Trajectory processing and correlation analysis were conducted using MDTraj 1.9.9 and NumPy 1.26.4 within a Conda-based Python 3.10 environment. Frames from the final 15% of each trajectory (750–1000 ns) were extracted, aligned to the initial minimized structure to remove global rotation and translation, and reduced to only Cα atoms.

Cα atomic displacements were reshaped into 2D matrices (frames × atoms × 3 coordinates), then flattened and mean-centered. Pearson correlation coefficients were computed using NumPy’s corrcoef function, yielding a symmetric residue–residue matrix per system. Δ-correlation matrices were generated by direct subtraction of the WT matrix from the mutant matrix. The most shifted residue pairs were identified by ranking the largest positive and negative Δ-correlation values. Visualization was performed with Matplotlib 3.8.4 and Seaborn 0.13.2. The script is available in the Appendix A.

## 5. Conclusions

The inhibitory activity of AT against the procoagulant protease FXa is tightly controlled by an allosteric mechanism triggered by heparin—one that has long remained incompletely understood. In this study, we identified and characterized an evolutionarily conserved ACN that bridges the HBS and RCL/exosite domains. This network functions as a molecular lock that stabilizes native AT in a reactivity-suppressed R state. Disruption of key residues within this network—particularly H120, Y131, and Y166—destabilized the R form and produced constitutively active variants of native AT, offering new mechanistic insights and a potential strategy for engineering heparin-independent AT therapeutics.

## Figures and Tables

**Figure 1 ijms-26-08984-f001:**
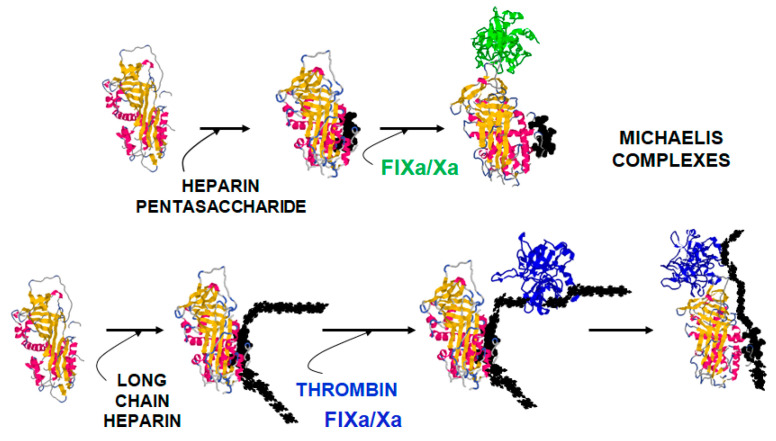
Formation of the transient antithrombin-protease Michaelis encounter complex. The heparin pentasaccharide allosterically activates antithrombin for the binding of factor IXa (FIXa) or factor Xa (FXa). Long heparin chains bridge the encounter complexes with all proteases.

**Figure 2 ijms-26-08984-f002:**
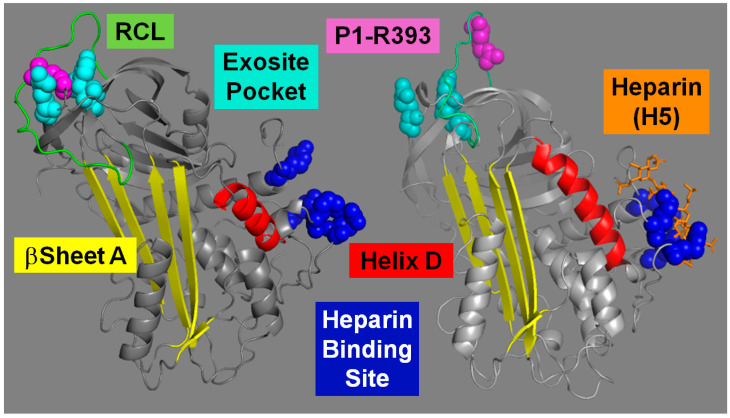
Structural differences between the native activity-silent and heparin-bound activated antithrombin forms. Crystal structures of the antithrombin’s native R form (PDB code 1T1F, **left panel**) and heparin-bound AH form (PDB code 1EO3, **right panel**). Some relevant protein domains for the present study are highlighted in colors. Heparin pentasaccharide (H5). The structure figures were produced using the software PyMOL 2.5.10.

**Figure 3 ijms-26-08984-f003:**
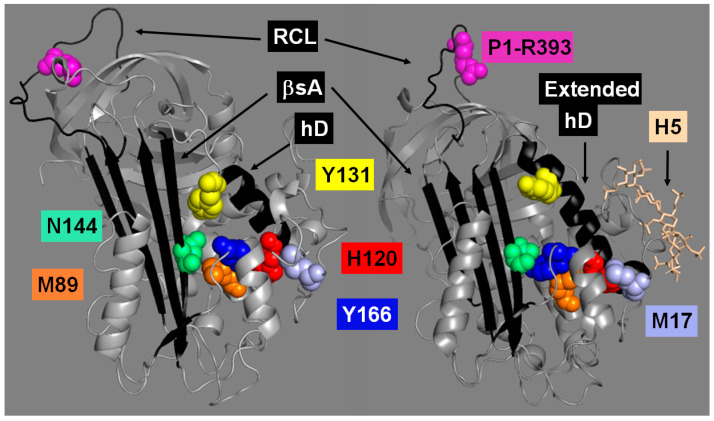
Antithrombin residues that undergo the largest displacements during the R-to-AH transition. Shown are the residues that had the largest displacements measured by our analysis that compared the structures of the antithrombin’s silenced (R-PDB code 1T1F—**left**) and heparin-activated (AH-PDB code 1EO3—**right**) forms. Structure representations were produced using the software PyMOL 2.5.10.

**Figure 4 ijms-26-08984-f004:**
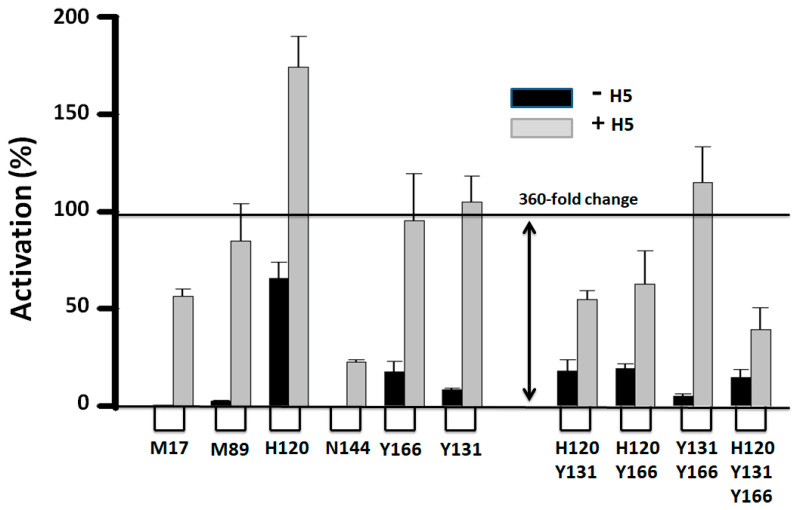
Site-directed mutational activation of antithrombin. The residues found to undergo the largest displacements during the R-to-AH transition were mutated. The effect of the mutations on reactivity with FXa was determined (Appendix A). The difference in reactivity between wild-type native and allosterically activated AT is 360-fold (100% activation). Shown are reactivities relative to the wild-type AT allosteric enhancement promoted by heparin pentasaccharide (H5). All single mutants were mutated into Val, except for Y131L. All double and triple mutations involved mutations into Leu. The plot was made using the software SigmaPlot 14.5.

**Figure 5 ijms-26-08984-f005:**
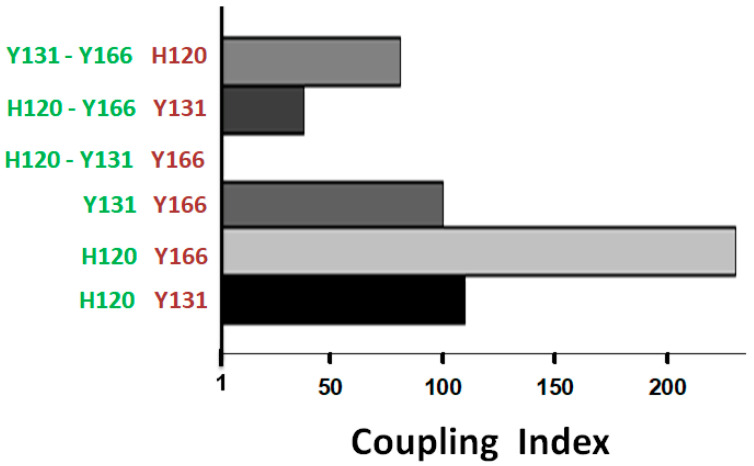
Thermodynamic linkages among ACN residues. Site-specific double-mutant thermodynamic coupling analysis was performed to probe the functional cooperative link between the residues in green versus the residues in purple (Appendix A). Coupling index values larger than 1 indicate cooperative linkage between the tested residues. All single mutants were mutated into Val, except for Y131L. All double and triple mutations involved mutations into Leu. The plot was made using the software SigmaPlot 14.5.

**Figure 6 ijms-26-08984-f006:**
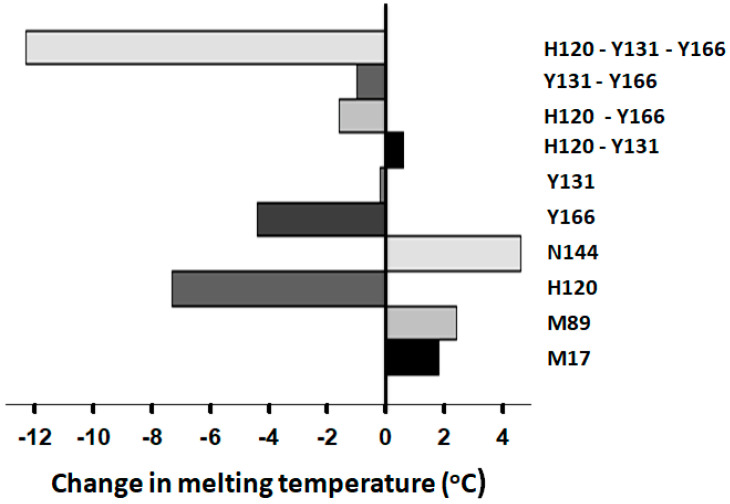
Mutations in the antithrombin ACN residues alter the protein melting temperature. Changes in the intrinsic tryptophan fluorescence as a dependence on temperature were used to measure protein thermal stability. Melting curves were computer fitted to the van Hoff equation to calculate T_m_ values (Appendix A). All single mutants were mutated into Val, except for Y131L. All double and triple mutations involved mutations into Leu. The plot was made using the software SigmaPlot 14.5.

**Figure 7 ijms-26-08984-f007:**
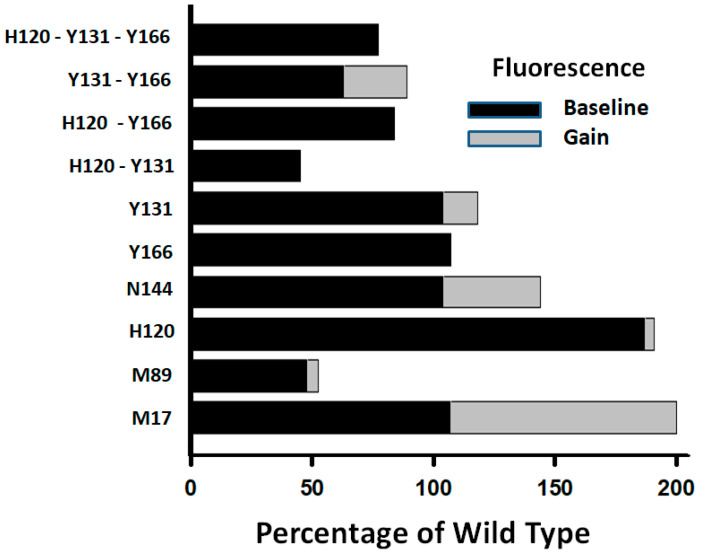
Protein fluorescence changes induced by mutations in the ACN residues report structural shifts. The baseline tryptophane fluorescence was measured as the linear dependence of the fluorescence on the antithrombin concentration. And the fluorescence gain was determined by adding heparin pentasaccharide at a saturating concentration five-fold larger than the AT concentration. Experimental values are shown in Appendix A. All single mutants were mutated into Val, except for Y131L. All double and triple mutations involved mutations into Leu. The plot was made using the software SigmaPlot 14.5.

**Figure 8 ijms-26-08984-f008:**
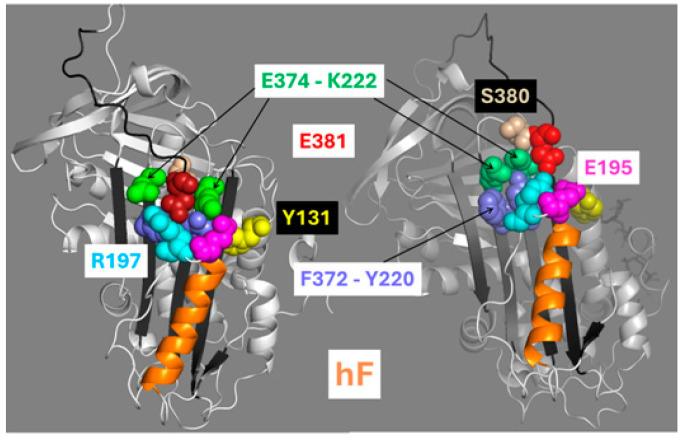
Residues in the antithrombin RCL hinge region that undergo the largest displacements during the R-to-AH transition. Crystal structures of the antithrombin’s forms R (PDB code 1T1F—**left**) and AH (PDB code 1EO3—**right**). Shown are residues with large displacements between these two forms (Appendix A). In the R form, S380 and E381 lie inserted between E374 and K222 (green). In the AH form, both RCL residues are expelled and the pairs E374-K222 and F372-Y220 reduce their gap separation. Structure representations were produced using the software PyMOL 2.5.10.

**Figure 9 ijms-26-08984-f009:**
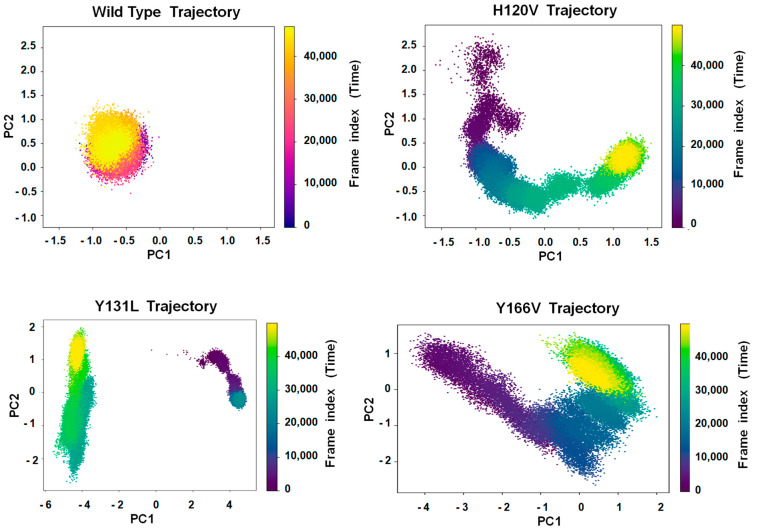
PCA temporal dynamics of wild-type and ACN mutation trajectories (Cα atoms). Com- parison of the principal component space explored by the protein structures over the course of their 0–1000 ns MDS. The Y131L simulation was split into 0–500 ns and 500–1000 ns trajectories. Each point represents a frame projected onto the first two principal components (PC1 and PC2), colored by frame index (i.e., simulation time).

**Figure 10 ijms-26-08984-f010:**
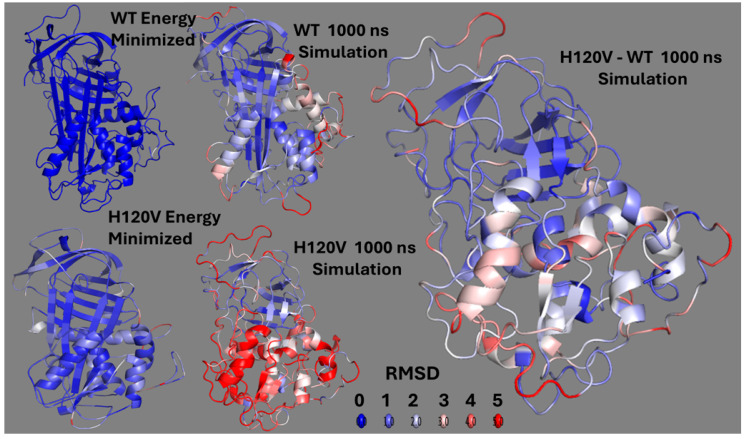
Structural impact of the H120V mutation on antithrombin dynamics and conformational deviation over 1000 ns of MDS. Representative structures are shown for wild-type (WT) and H120V antithrombin at the energy-minimized state and after 1000 ns of MDS. Structures are colored by per-atom RMSD as heatmaps and were visualized in PyMOL 2.5.10 using a custom script that assigns B-factors proportional to atomic displacement and applies a blue-to-red spectrum ranging from 0 to 5 Å, with blue indicating minimal deviation and red highlighting regions of highest divergence. **Top left:** Energy-minimized WT structure compared to the pre-minimized state. **Top center:** Final WT conformation after 1000 ns of simulation colored relative to the WT minimized structure used as reference. **Bottom left:** Energy-minimized structure of the H120V mutant compared to its pre-minimized state. **Bottom center:** Final mutant conformation after 1000 ns colored relative to the mutant minimized structure. **Right panel:** Differential RMSD heatmap between the final mutant and WT structures.

**Figure 11 ijms-26-08984-f011:**
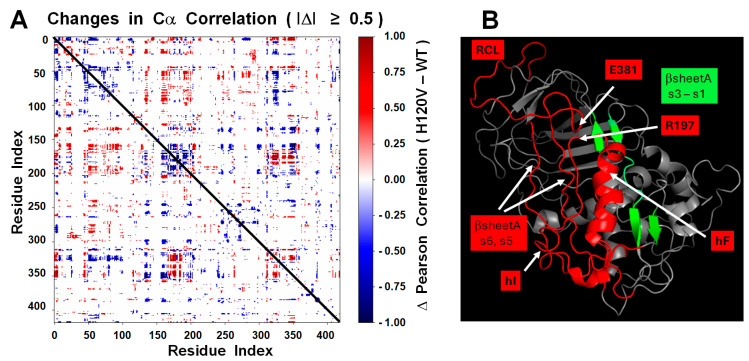
Differential Cα correlation and structural mapping of allosteric effects induced by the H120V mutation. (**A**) Differential Pearson correlation heatmap (Δρ) of residue-wise Cα displacements between the H120V and wild-type (WT) simulations. The matrix was computed by first aligning all trajectory frames to the starting frame using backbone atoms and then calculating the Pearson correlation coefficient of the positional fluctuations between all pairs of Cα atoms. Each matrix element represents the change in dynamic correlation for a given residue pair (i, j), defined as Δρ = ρ_H120V–ρ_WT. Only changes with absolute magnitude |Δρ| ≥ 0.5 are visualized. Positive Δρ values (red) indicate increased dynamic coupling in the mutant, while negative values (blue) indicate weakened or lost correlations. The matrix is symmetrical and highlights both local and distal correlation shifts potentially involved in allosteric signaling pathways altered by the mutation. (**B**) Structural mapping of the key protein regions that were involved in significant increments, and decrements in correlation changes between the H120V and WT simulations (Appendix A). The structure is that of the H120V mutant after 1000 ns MDS. It is relevant to notice that the entire length of strands 5 and 6, and partial lengths of strands 2 and 3 of βsheet A, as well as hI have lost their secondary structure to become loop-like structures. The structure was generated using the software PyMol 2.5.10.

**Figure 12 ijms-26-08984-f012:**
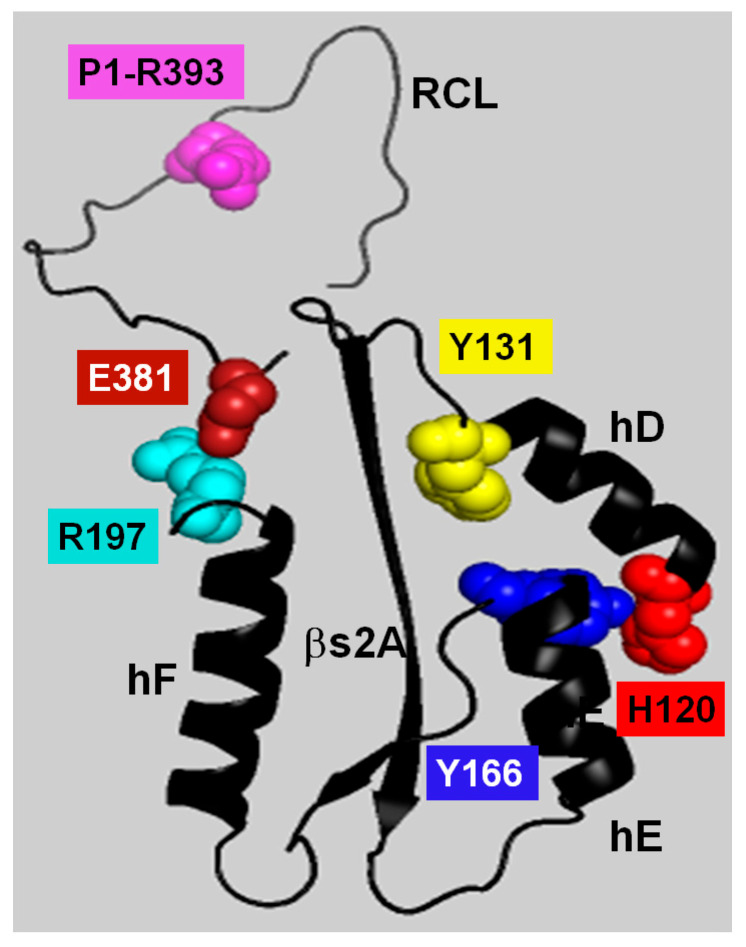
Structural representation of the antithrombin allosteric communication network (ACN). The ACN connects the heparin-binding site (HBS) to the reactive center loop (RCL). The figure shows the residues in the native repressed (R) state. Key residues in the network are shown as space-filling spheres: H120 and Y131 in helix D, and Y166 in helix E. Also highlighted are the RCL hinge residue E381 and its interacting partner R197, which is located at the loop after helix F. The P1-R393 residue is shown in the RCL. Secondary structure elements are colored black (α-helices D, E, F; β-strand s2A; and the RCL). This schematic highlights the spatial arrangement of ACN residues between the HBS and RCL that mediate protein stabilization and activation during the R-to-AH conformational transition.

## Data Availability

The data presented in this study are available on request from the corresponding author.

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
