# Peer review of "The Allosteric Communication Network in the Activation of Antithrombin by Heparin"

_ijms, 2025, doi:10.3390/ijms26188984_

Round 1
Reviewer 1 Report
Comments and Suggestions for Authors
Attached in PDF

Author Response
Thanks

Reviewer 2 Report
Comments and Suggestions for Authors
The author uses computational techniques based on atomistic models to dissect the mechanism of communication between heparin-binding site (HBS) and reactive center loop (RCL) in antithrombin protein.
The model well describes the silenced nature of inactive (R) state, contrasted to the features of an activated state when heparin is bound (A*).
Release of silencing by changing target sites shows the importance of the design of silencing.
In vitro experiments (reactivity with FXa, thermal stability) were used to address hot spots in allosteric communication network (ACN) modifications.
The work reported in the manuscript is a more detailed description of results reported by previous works where the same author was involved (refs. 7-9, 12).
A similar mechanism has been proposed for serpins in general (ref. 11 and others cited).
The data discussed in this manuscript provide information useful to clear the network of interactions in the R-to-AH transition.
RCL activity was measured using thrombin and FXa as monitored by Trp fluorescence.
The manuscript is of wide potential interest.
However, conclusions are weakly supported by results.
This is why I recommend a major revision of this work.
The main missing part are numerical simulations, that are in a sort of preliminary step.
I recommend the author to extend the molecular dynamics simulations (MDS) to a wider pool of variants, as it is usually done in similar cases.
This will increase the statistics and, therefore, reliability of conclusions.
As for experiments, one important missing part is the analysis of thermal stability via circular dichroism (CD).
Some details are discussed below.
P.2
Figure 1 - Indicate in the caption the references of crystal structures used to draw both panels (including PDB codes).
L.53 - Expand P1 the first time it is used (as the other many acronyms).
The same holds for other Px (Sect. 2.5, P.7, etc.).
L.68 - This paragraph describes the original contribution reported in the manuscript:
the author should explicitly tell the reader, it is not clear.
Sect. 2.1 / 4.1
This analysis is severely limited by the available crystal information.
To compare the two structures the wobbling of each of them should be taken into account.
This is usually done by analyzing molecular dynamics simulations started from experimental structures.
Sect. 2.2
P.4
Figure 3 and related discussion (L.104-125) - It is not clear when mutation is to Val or Leu.
The author should indicate the final residue in each displayed mutation experiment.
L. 148-150 - The sentence is not clear.
"inhibitory ACN" would mean an allosteric communication network performing inhibition, but it is not clear inhibition of what.
The reader understands inhibition of AT reactivity, but one can also understand that the inhibited function is "repression of reactivity" (silencing).
Please, clarify.
Sect. 2.4
L.156-166 - According to this report, there is no clear correlation between Delta Tm (change in Tm) and extent of activation.
Tm changes are usually measured by changes in CD spectra (Theta at lambda=222 nm).
Trp fluorescence is more sensitive to local environment.
A clearer picture is obtained by collecting both data.
Sect. 2.5
P.7
L.195-215 - It is not clear which is the supporting evidence of the sentences:
is the deviation measured in Sect. 2.1?
L.206 - E381G mutation was not mentioned before (Sects. 2.2 and related Methods).
Sect. 2.6
P.8
L.251-253 - There is no evidence here of "concerted communication".
The RMSD of backbone atoms in the 2 simulations should be provided (maybe in SI) as a function of time, to understand if in WT the chosen configuration, representative of R AT state, is at equilibrium in water solution.
Reading Sect. 4.6 we understand that no ions (NaCl) were added.
This approximation is expected to dramatically destabilize the protein, being the latter highly charged.
Also, AT is charged because of modulation by heparin (and Ca2+, indeed).
The reliability of L.235-237 (critical role of E381-R197 salt-bridge) is affected by the approximation of protein dielectric medium.
In any case, these kind of simulations is useful when different mutations are compared and electrostatics is important.
For instance, it would be interesting to compare the structural changes occurring in H120V (where Tm decreases) with those of M17V and M89V (where Tm increases).
In theory, the simulation allows to measue the RMSD of backbone with respet to R state (PDB 1T1F) together with RMSD with respect to AH state (PDB 1EO3).
In H120V the second RMSD should change less than the first one, or even decrease.
The structure in Ref.21 (AH) can be also used to probe if AT variants can differently prepare the protein scaffold to heparin binding (pre-organization).
Simulations (MDS) of all investigated mutant have the potential to assess sentence at L.325-326.
Unfortunately, the study by MDS of many variants is imperative.
The latter approach is the one usually applied to this study.
Sect. 4.2
P.14
L.481 - "ACN mutant proteins": ACN is not the protein.
Discussion
P.9
Figure 9, L.287-288 - The caption can not be understood.
Figure S2 - It is not clear where are the structures used to draw panels from.
In particular a reference to the ternary long-chain heparin/AT/FIXa complex drawn bottom-right is required.
Author Response
Thanks

Round 2
Reviewer 1 Report
Comments and Suggestions for Authors
I thank Gonzalo Izaguirre for having accepted my suggestions to improve his paper. In my opinion, the new version he submitted is adequate for publication.
Author Response
I thank Gonzalo Izaguirre for having accepted my suggestions to improve his paper. In my opinion, the new version he submitted is adequate for publication.
Thanks you for your help.
Reviewer 2 Report
Comments and Suggestions for Authors
The author clarified some parts of the discussion and extended the two numerical simulations performed, H120V variant and the wild-type sequence. They applied a correlation measure to assess convergence of the two simulations.
Therefore the major statistical limitation that I pointed out, in terms of sequence variants, persists in the presented results, while at the same computational cost one more variant would be feasible.
I do not see improvements in statistics.
Author Response
The author clarified some parts of the discussion and extended the two numerical simulations performed, H120V variant and the wild-type sequence. They applied a correlation measure to assess convergence of the two simulations.
The discussion has been further improved to include the results from the two additional simulations for mutants Y131L and Y166V.
Therefore, the major statistical limitation that I pointed out, in terms of sequence variants, persists in the presented results, while at the same computational cost one more variant would be feasible.
Two more mutant simulations were added to provide a larger representation.
I do not see improvements in statistics.
I was referring to the PCA and Pearson analyses which are statistical in nature.
Thanks much for your help
Round 3
Reviewer 2 Report
Comments and Suggestions for Authors In the 3rd revision the author added two variants to the simulations. The added results are interesting, showing the different evolution of Y131L with respcet to H120V and Y166V (Fig.9 and related comments L.272-281), the latter two variants displaying thermal destabilization compared to WT (Fig.6). The comments to the added simulations (Y131L and Y166V) are limited to L. 272-281. The similar H120V and Y131L dominance by a "single, strong coherent conformational flutcuation" conflicts with Fig.9, where Y166V and H120V look similar. Since Y131L variant displays DTm about zero (Fig.6), a similarity between Y131L and WT looks more consistent with experimental data. I recommend the author to clarify a possible consistency between this section and DTm. Or, conversely, to explain why there is a discrepancy.Author Response
In the 3rd revision the author added two variants to the simulations. The added results are interesting, showing the different evolution of Y131L with respcet to H120V and Y166V (Fig.9 and related comments L.272-281), the latter two variants displaying thermal destabilization compared to WT (Fig.6). The comments to the added simulations (Y131L and Y166V) are limited to L. 272-281. The similar H120V and Y131L dominance by a "single, strong coherent conformational flutcuation" conflicts with Fig.9, where Y166V and H120V look similar. Since Y131L variant displays DTm about zero (Fig.6), a similarity between Y131L and WT looks more consistent with experimental data. I recommend the author to clarify a possible consistency between this section and DTm. Or, conversely, to explain why there is a discrepancy.
The paragraph that the reviewer is referring to compares the distribution of variance among the four simulations. WT and Y166V simulations had a wider spread of variance among PCs, which indicates that these simulations underwent multi-modal trajectories. On the other hand, variance for the H120V and Y131L mutants concentrated into PC1, which indicates that these simulations followed mostly a single trajectory. These differences do not relate to the stability of the protein. A highly unstable protein could follow single or multiple paths during simulations.